

# Furthering information from OH and HO$_2$+RO$_2$ observations using a high resolution time of flight mass spectrometer

R.L. Mauldin III[1,2] , M.P. Rissanen[1], T. Petäjä[1], and M. Kulmala[1]

[1] Department of Physics, University of Helsinki, P.O. Box 64, 00014, Finland

[2] Department of Atmospheric and Oceanic Sciences, University of Colorado, Boulder, Colorado

*Correspondence to*: R.L. Mauldin (roy.mauldin@colorado.edu)

**Abstract.** An instrument has been developed which allows an entire suite of oxidation products (H$_2$SO$_4$ and Extremely Low Volatile Organic Compounds, ELVOCs) to be measured along with the concentrations of the key radical species OH and

HO$_2$+RO$_2$. The system combines the techniques for chemical conversion of OH or HO$_2$+RO$_2$ into H$_2$SO$_4$ together with nitrate ion (NO$_3^-$) Chemical Ionization (CI) and Atmospheric Pressure interface - Time of Flight Mass Spectrometer (NO$_3^-$ CI-APi-ToF) which has been previously used for the detection of ELVOCs and a few other oxygenated organic species. The system exhibits the same sensitivity towards OH or HO$_2$+RO$_2$ as previous quadrupole chemical ionization (CIMS) measurements with limits of detection of ~2 x 10$^5$ and ~2 x 10$^6$ molecule cm$^{-3}$ for OH and HO$_2$+RO$_2$ respectively. Unlike

previous CIMS measurements, the use of a NO$_3^-$ CI-APi-ToF allows the acquisition of the entire mass spectrum at high resolution with each measurement, allowing one to see how the organic species behave when the sample flow is perturbed with reagent gasses (SO$_2$ or NO). While the combination of these measurements into one instrument is of great practical value, it is the combination of these data within the spectra obtained in each mode that extends the information far beyond the measurements themselves.

## 1 Introduction

Oxidation is a central atmospheric process and impacts such issues as climate change, air quality, and acid rain. Key here are reactions involving the hydroxyl radical (OH), the hydroperoxy radical (HO$_2$), and organic peroxy radicals (RO$_2$). OH reacts with volatile organic compounds (VOCs) usually leading to the formation of organic peroxy radicals, which in the presence of nitric oxide (NO) primarily regenerate OH. This fast cycling of radicals, which results in chemical lifetimes of OH less

than one second, controls many aspects of atmospheric chemistry such as the formation of ozone and secondary organic aerosols, and the removal of methane and other greenhouse gases such as halocarbons that affect the radiative balance of the atmosphere. Because of this central role, measurements of these radicals can provide a critical test of our understanding of the fast photochemistry of the atmosphere, as the concentration of OH in the atmosphere is determined by the chemistry of VOCs and NO$_x$ rather than the result of transport (Heard and Pilling, 2003).





Towards these ends, two techniques have emerged as the most commonly used for the detection of peroxy radicals and OH. One technique, often referred to as FAGE (Fluorescence Assay by Gas Expansion), involves the laser induced fluorescence, LIF, detection of OH. $HO_2$ is measured by first converting it to OH which is subsequently detected. The other technique, utilized in this work, involves the addition of reagent gases to the sample flow causing the chemical conversion of the radical

species into sulfuric acid, $H_2SO_4$, which is subsequently detected via nitrate ion ($NO_3^-$) chemical ionization mass spectroscopy, CIMS. By changing the species or concentration of the added reagents (NO, $SO_2$, propane) it is possible to measure ambient concentrations of OH, $HO_2$, and $HO_2+RO_2$, as well as ambient $H_2SO_4$.

While the $NO_3^-$ ion CIMS has been in use for decades, the limitations of a quadrupole mass spectrometer have mainly limited its use to the measurement of $H_2SO_4$. Recently, $NO_3^-$ ion chemical ionization has been coupled to an APi-TOF mass

spectrometer and is proving to be a highly valuable measurement on its own. Jokinen et al. (2012) have used the $NO_3^-$ CI-APi-ToF technique to measure sulfuric acid clusters. It has since been used it to examine the formation of extremely low volatility organic, or ELVOC, species (Ehn et al. ,2014; Jokinen et al., 2014; Rissanen et al. 2014, 2015).

Here we describe a single instrument capable of measuring OH, $HO_2+RO_2$, $H_2SO_4$ and other species detectable by nitrate chemical ionization. It employs two modes where reagent gasses are added to the sample, for the separate determination of

OH and $HO_2+RO_2$, and third $NO_3^-$ ion mode where no reagent gases are added. While the use of one instrument to perform these measurements is practical, the fact that the spectra are all obtained with the same instrumental constraints (calibration, resolution, transmission, etc.) allows for accurate intercomparison of spectra obtained in each mode. Observing the behavior of the spectrum as whole under the varying conditions where reagent gasses are added or not, chemical data can be obtained towards the identity of many of the organic peaks observed, thus furthering the information gained beyond the individual

OH, and $HO_2+RO_2$ measurements by themselves.

## 2 Instrumentation

A schematic of the instrument is shown in Figure 1. The instrument combines the techniques of the CIMS OH technique developed by Eisele et al. (1991, 1993) where $SO_2$ is added to the sample flow, and the $HO_2+RO_2$ technique described by Edwards et al. (2003), where $SO_2$ and NO are added to the sample flow, with the additional extension of High Resolution

Time of Flight mass detection. Briefly, the instrument can be operated in three different modes: One for the detection of hydroxyl radicals (OH mode), one for the detection of peroxy radicals ($HO_2+RO_2$ mode), and one where the sample flow is not treated with reagent compounds ($NO_3^-$ mode). These modes allow the sequential determination of [OH], [$HO_2+RO_2$], [$H_2SO_4$], an upper limit to the concentration of other $SO_2$ oxidants as well as numerous compounds which have been previously identified as Extremely Low Volatility Organic Compounds or ELVOCs. The combination of OH and $HO_2+RO_2$

measurements has been performed before using a quadrupole mass spectrometer (Sjostedt et al., 2007; Mauldin et al., 2004). The $HO_xRO_x$ instrument described here extends these abilities with the addition of the $NO_3^-$ ion mode and the use of HR ToF detection. To the author's knowledge, this is the first use of an HR ToF for either OH or $HO_2+RO_2$ measurements. As will be





pointed out below, the combination of these features allows more information to be gained than just the measurement of the individual concentrations.

Measurements were performed during April-May 2014 at the Hyytiälä research station located in the boreal forest approximately 200 km north of Helsinki, Finland. The instrument was located in a trailer with the inlet extending ~30 cm

beyond the southern facing wall. Ambient air was drawn in via a 10.1 cm ID inlet with a flow velocity of ~2.5 m s$^{-1}$. Air is sampled from the central portion of this flow into the instrument via a 1.9 cm ID OD stainless steel tube (wall thickness ~0.01 cm). Depending upon the measurement mode, reagent compounds are added to this sample flow before entering the ion reaction region (see Figure 1). Details of the chemistry of each mode are given below.

## 2.1 $NO_3^-$ Mode

The $NO_3^-$ mode is the simplest operating mode where no reagent gasses are added to the sample flow. This mode also forms the basis of detection and quantification of $H_2SO_4$ (detected as $HSO_4^-$ and $HNO_3 \cdot HSO_4^-$) formed from the other modes. The technique has been described previously for the use with a quadrupole mass spectrometer (Eisele et al., 1991, 1993; Mauldin et al., 1998; Petäjä et al., 2009) and more recently with a High Resolution Time of Flight mass spectrometer (Jokinen et al., 2012). Briefly, air flow was sampled into the ionization region via a 1.9 cm ID stainless steel sampling tube. Inside the

ionization region, a flow of air is introduced which surrounds the stainless steel sampling tube. This flow of air, termed the sheath flow, had been filtered to remove any $SO_2$ and "spiked" with a small amount (~$10^{11}$ molecule cm$^{-3}$) of $HNO_3$. Upon entering the ionization region, the outer coaxial layer of sheath air was ionized by passing it over a 0.635 cm wide piece of $^{241}$Am foil mounted on a 3.8 cm ID stainless steel tube concentric with the 1.9 cm tube. The $^{241}$Am alpha source (~5 MeV) was positioned so as to minimize ionization of the sample flow, and ensure that the initial ionization remain confined to the

outer sheath flow region. As the sheath flow passes over the $^{241}$Am source, the added $HNO_3$ is ionized in a multi-step process to form $NO_3^- \cdot (HNO_3)_n$ reagent ions (n=0,1,2). The $NO_3^- \cdot HNO_3$ ions are used as the primary reagent ion instead of $NO_3^-$ or $NO_3^- \cdot (HNO_3)_2$ because the sampled air can have a sufficient concentration of $HNO_3$ to convert $NO_3^-$ formed in the ion source into $NO_3^- \cdot HNO_3$ (Tanner et al., 1997). Exiting the ion source region the $NO_3^- \cdot (HNO_3)_n$ ions, but not the gas in which they were formed, are directed by means of electrostatic lenses through the intermediate, un-ionized, annular layer of

sheath gas into the central sample flow. Once in the sampled air flow, the $NO_3^- \cdot (HNO_3)_n$ reagent ions react with more acidic species, such as $H_2SO_4$, to produce product ions:

$$H_2SO_4 \ + \ NO_3^- \ \rightarrow \ HSO_4^- \ + \ HNO_3 \tag{R1}$$

Alternatively, the $NO_3^-$ ions can cluster with certain other species. The clustering reaction is particularly sensitive for highly oxidized compounds containing –OOH and -OH groups capable of forming hydrogen bonds with the nitrate ion (Rissanen et

al., 2014; Hyttinen et al., 2015). For clarity, only the core ion species are shown in the above equation. At the end of the ion reaction region the reagent and product ions are directed towards the vacuum system entrance by means of electrostatic fields.



## 2.2 OH Mode

The technique used here for the measurement of OH has been previously described (Eisele and Tanner, 1991; Eisele and Tanner, 1993; Tanner et al., 1997, Mauldin et al., 1998, Berresheim et al., 2000). Briefly, as the sample air flow is drawn in through the 1.9 cm inlet, a small amount ($\sim 3 \times 10^{13}$ molecule cm$^{-3}$) of SO$_2$ is added to the flow through a pair of 0.023 cm ID

transversely opposed injectors (Injector Pair #2) located near the front opening. The OH is then converted into H$_2$SO$_4$ via the reaction sequence:

$$OH \; + \; SO_2 \; + \; M \; \rightarrow \; HSO_3 \; + \; M \qquad\qquad\qquad (R2)$$

$$HSO_3 \; + \; O_2 \; \rightarrow \; SO_3 \; + \; HO_2 \qquad\qquad\qquad\qquad (R3)$$

$$SO_3 \; + \; 2H_2O \; + \; M \; \rightarrow \; H_2SO_4 \; + \; H_2O \; + \; M \qquad\qquad (R4)$$

Previous quadrupole CIMS measurements of OH have incorporated the use of isotopically labeled $^{34}$SO$_2$ to allow the simultaneous measurement of naturally occurring H$_2$SO$_4$ as well as OH. Here, due to cost, unlabeled SO$_2$ was used as the reagent. The shortcoming of the use of the unlabeled SO$_2$ is the inability to measure ambient H$_2$SO$_4$ simultaneously while in the OH measurement mode. However, ambient H$_2$SO$_4$ concentrations are separately determined while in the NO$_3^-$ mode. To prevent cycling of HO$_2$ and RO$_2$ back into OH, propane (an OH scavenger) is also added at much higher concentrations on a

continuous basis through a second pair of injectors (Injector Pair #3) located $\sim 5$ cm downstream from the SO$_2$ injection. This propane removes any OH which has been cycled back from HO$_2$ and RO$_2$ via reactions with NO or O$_3$ after the ambient OH initially present has been converted into H$_2$SO$_4$. Once formed, the H$_2$SO$_4$ is detected via its product ion at masses 97 and 160 Th. To account for other non-OH oxidation processes that can also lead to H$_2$SO$_4$ formation from SO$_2$, a second measurement is made where $\sim 1 \times 10^{16}$ molecule cm$^{-3}$ of propane is added together with the SO$_2$ through injector pair #2. This

concentration is sufficient to scavenge >98% of the OH present via reaction with propane instead of SO$_2$. Hence, the concentration of OH is attributed to the difference between the total SO$_2$ oxidant signal (that with only SO$_2$ added through injector pair #2) and the non-OH SO$_2$ oxidant signal (with both propane and SO$_2$ added through injector pair #2). The source of the non-OH SO$_2$ oxidant signal has been found to follow the same characteristics as stabilized Criegee intermediates resulting from the ozonolysis of biogenic alkenes (Mauldin et al., 2012). Thus, this signal can be used to give an estimate of

Criegee radical concentrations. However, contributions from other non-OH, non-Criegee species cannot be ruled out.

## 2.3 HO$_2$+RO$_2$ Mode

The measurement of HO$_2$+RO$_2$ incorporates the same technique as described by Cantrell et al., (2003) and Mauldin et al., (2004). Briefly, the technique involves the chemical conversion of peroxy radicals to H$_2$SO$_4$. As sampled air was drawn through the 1.9 cm ID inlet, a mixture of NO and SO$_2$ was added to the flow via a pair of injectors (Injector Pair #1).

Conversion of HO$_2$+RO$_2$ to H$_2$SO$_4$ then takes place via the following reaction sequence:

$$RO_2 \; + \; NO \; \rightarrow \; RO \; + \; NO_2 \qquad\qquad\qquad\qquad (R5)$$

$$RO_2 \; + \; NO \; \rightarrow \; RONO_2 \qquad\qquad\qquad\qquad (R5a)$$


$$RO + O_2 \rightarrow R'CHO + HO_2 \tag{R6}$$

$$HO_2 + NO \rightarrow OH + NO_2 \tag{R7}$$

$$OH + SO_2 + M \rightarrow HOSO_2 + M \tag{R2}$$

$$HOSO_2 + O_2 \rightarrow HO_2 + SO_3 \tag{R3}$$

$$SO_3 + 2H_2O \rightarrow H_2SO_4 + H_2O \tag{R4}$$

$$OH + NO + M \rightarrow HONO + M \tag{R8}$$

It is quite likely that some of the RO radicals produced in reaction R5 do not produce $HO_2$ from their reaction with $O_2$, but do decompose to form measurable radicals (Cantrell et al., 2003). These radicals are rapidly transformed back to corresponding $RO_2$ radicals and thus may contribute doubly to the measured $[HO_2+RO_2]$. This increase however, is not expected to be particularly significant, as it is largely cancelled by reactions forming organo-nitrates through reaction 4a. This formation of organo-nitrates is especially important for the larger $RO_2$ radicals produced from terpene oxidation (Jokinen et al., 2014).

The above reaction sequence represents a chain reaction producing $H_2SO_4$, with OH reacting with either $SO_2$ in reaction R(2) to propagate the chain, or with NO in reaction R(8) which terminates the chain. The yield of $H_2SO_4$ per $HO_2$ is determined by the reaction time and the $[SO_2]/[NO]$ ratio (Cantrell et al., 2003). Larger $[SO_2]/[NO]$ ratios leads to a larger gain or increase in the concentration of $H_2SO_4$ per $HO_2$. In the current study the $[SO_2]/[NO]$ ratio was adjusted such that the gain of the system was ~1, i.e., each $HO_2$ produced, on average, one $H_2SO_4$ molecule. In the present study the $SO_2$ and NO concentrations were ~300 ppbv and 50 ppbv, respectively. At still higher concentrations of NO (>1000 ppmv), the reaction of RO radicals via (2) is inhibited by the reaction:

$$RO + NO + M \rightarrow RONO + M \tag{R9}$$

This reaction has been exploited for purposes of measuring $HO_2$ only (not $HO_2+RO_2$) (Cantrell et al., 2003; Brune et al., 1995). In the current study, however, technical constraints prevented the use of such high concentrations of NO, limiting the measurements to only $HO_2+RO_2$.

As in the case of the OH measurement, a second measurement is also performed to account for $H_2SO_4$ not produced from the conversion of $HO_2$. Here NO is added first (Injector Pair #1), and $SO_2$ is added 50 msec downstream of the NO addition (Injector Pair #2). NO is added in sufficient concentration such that 99% of the OH formed is converted to HONO before the addition of the $SO_2$. To prevent any interference from NO in the OH or $NO_3^-$ modes, the flow of the front pair of injectors (Injector Pair #1) is reversed to prevent any residual reagent gases from leaking into the sample flow.

## 2.4 Calibration

Two different methods are used to calibrate the system, both of which rely on water photolysis at 185 nm. In the field, calibration of the system is performed by producing a known amount of OH in front of the 1.9 cm sampling port (Tanner and Eisele, 1995; Tanner et al., 1997). This photolysis produces both OH and $HO_2$ via:





$$H_2O + h\nu\,(184.9\,nm) \rightarrow OH + H \qquad\qquad (R10)$$

$$H + O_2 + M \rightarrow HO_2 + M \qquad\qquad (R11)$$

Light from a temperature controlled Hg lamp is reflected by two mirrors coated to selectively reflect the 184.9 nm emission line. The OH concentration produced by the calibration source is dependent upon the intensity (photo flux) at 184.9 nm, the

$[H_2O]$, the cross-section of $H_2O$ at 184.9 nm, the yield of OH from $H_2O$ photolysis, and the sample flow velocity. The flow velocity in the 10.1 cm duct was measured using a hotwire anemometer and the $[H_2O]$ was measured by a hygrometer. A value of $7.2 \times 10^{-20}$ cm$^2$ molecule$^{-1}$ was used for the absorption cross-section of $H_2O$ at 184.9 nm (Cantrell et al., 1997), and the OH yield from the photolysis at 184.9 nm is ~100% (Baulch et al., 1982). Photon flux at 184.9 nm was mapped out using a vacuum UV diode mounted on an x/y movable drive. This diode was intercompared with a National Institute of

Standards and Technology (NIST) standard diode before and after the study.

Calibration of the $HO_2+RO_2$ system was performed using the same system as for OH and $H_2SO_4$. Again, the amount of $HO_2$ produced is a function of the ambient $H_2O$, the Hg Pen Ray lamp flux at 184.9 nm, and the sample flow velocity. As both the OH and $HO_2$ are converted into $H_2SO_4$ via reactions (2-7), when in the HO2+RO2 mode, the resulting calibration coefficient is for the sum of OH and $HO_2$. If, therefore, the gain of the system is ~1, the calibration value should be approximately twice

that for OH and $H_2SO_4$. Hence, this determination serves as both a calibration of the $HO_2+RO_2$ measurement as well as the gain of the system. The average calibration coefficient for $HO_2+RO_2$ obtained during the study was $(5.63\pm0.67) \times 10^9$ molecule cm$^{-3}$, and the coefficient for OH and $H_2SO_4$ was $(5.24\pm0.46) \times 10^9$ molecule cm$^{-3}$. These values indicate that the gain of the system was very close to one during study. It should be noted that these values are comparable to those previously obtained using a quadrupole system (Petäjä et al., 2009) indicating that the chemical conversion/ion reaction

system was performing in a similar manner.

While not used in the field for this study, the OH and $H_2SO_4$ measurement can also be calibrated by a system that uses $H_2O$ photolysis in presence of $SO_2$, to produce a known amount $H_2SO_4$ (Kürtén et al., 2012). In the past, this calibration system has produced similar values for the calibration coefficient (5-6) $\times 10^9$ molecule cm$^{-3}$ for a similar inlet configuration.

Using this calibration coefficient, the calculated limit of detection for the OH measurement is ~2 $\times 10^5$ molecule cm$^{-3}$, and ~2

$\times 10^6$ molecule cm$^{-3}$ for $(HO_2+RO_2)$. The uncertainty in the current set of measurements is estimated at $\pm$ 40% for OH and $H_2SO_4$. For $HO_2 + RO_2$ the uncertainty is somewhat larger, being $\pm$ 60%. The error value is 2□ and is calculated from a propagation of errors calculation which includes both the total systematic and random errors for a given measurement.

## 3 Results and Discussion

Figure 2 shows a typical mass spectrum obtained from the $HO_xRO_x$ instrument at the Hyytiälä research station. The major

peaks have been labeled with their mass and the compounds they represent. The spectrum was obtained while the instrument was in the $HO_2+RO_2$ "signal" mode to better view the $HSO_4^-$ (mass 97) and $(H_2SO_4)NO_3^-$ (mass 160) product peaks. The high resolution of the HR-ToF allows these product ions to be detected at their exact masses, 96.9601 amu and 159.9557





amu ensuring that they indeed were $HSO_4^-$ and $(H_2SO_4)NO_3^-$ respectively. It can be seen that the reagent nitrate ion peaks; $NO_3^-$, $(HNO_3)NO_3^-$, and $(HNO_3)_2NO_3^-$, are the dominant peaks in the spectra. Even though the spectrum shown in Figure 2 was obtained in the $HO_2+RO_2$ "signal" mode which produces the most product $H_2SO_4$, the reagent ion still comprises ~97% of the total ion signal.

For the data presented here the instrument was run sequentially through the $NO_3^-$ ion, OH, and $HO_2+RO_2$ modes. A sample trace of the measured $H_2SO_4$ concentration versus time for the three different measurement modes is shown in Figure 3. The data presented are un-averaged one second measurements, so that the response of the instrument to changes in modes can be clearly seen. At 13:08:30, the sequence begins with the instrument transitioning into the $NO_3^-$ ion mode. As this transition simply involves switching a valve to draw ~100 sccm of flow from injector pair #1 and turning off the added reagent gases,

it should be relatively fast. From the plot, it can be seen the $H_2SO_4$ concentration drops and stabilizes over the period of a 2-3 seconds to an average concentration of ~3.5 x $10^6$ molecule cm$^{-3}$. The instrument remains in this mode for 240 sec. At ~13:12:30, the $SO_2$ and propane flows are then turned on and sent to injector pairs #2 and #3 respectively. This configuration is termed the OH "signal" mode, and the instrument remains in this mode for 30 sec. At this point propane is also added along with the $SO_2$ through injector pair #2 putting the instrument into the OH "background" mode. After 30 sec in this

mode the instrument switches back to the OH "signal" mode. This switching between the OH "signal" and "background" modes continues for 240 sec allowing for four OH "signal" and "background" measurements to be obtained. Lines indicating the average $H_2SO_4$ concentration are also shown in the figure. The difference between the OH "signal" and "background" values is the concentration of $H_2SO_4$ attribute to production from OH. As can be seen from the figure, the $SO_2$ flow had not equilibrated for the first "signal" measurement and which is thus not used. From the data shown in Figure 3, the

concentration of OH is ~3 x $10^6$ molecule cm$^{-3}$. At this point propane is turned off, the NO flow turned on, the $SO_2$ flow adjusted, and valves switch such that NO and $SO_2$ are added through injector pair #1 putting the instrument into what is termed the $HO_2+RO_2$ "signal" mode. This instrument remains in this configuration for 30 sec at which point the $SO_2$ flow is switched so that it added through injector pair #2, putting the instrument into the $HO_2+RO_2$ "background" mode. After 30 sec in this mode the instrument switches back to the $HO_2+RO_2$ "signal" mode. This switching between the $HO_2+RO_2$

"signal" and "background" modes continues for 240 sec allowing for four $HO_2+RO_2$ "signal" and "background" measurements to be obtained. The difference between the $HO_2+RO_2$ "signal" and "background" values is the concentration of $H_2SO_4$ attribute to production from peroxy radicals. As can be seen from the figure, the $SO_2$ flow had not equilibrated for the first "signal" measurement and which is thus not used. From the data shown in Figure 3, the concentration of $HO_2+RO_2$ is (7-8) x $10^8$ molecule cm$^{-3}$. At this point the measurement sequence is restarted by turning off all added reagents putting the

instrument back into the $NO_3^-$ ion mode. Using this measurement scheme, four 1 minute measurements of $HO_2+RO_2$, four 1 minute measurements of OH, and 240 sec of $NO_3^-$ ion mode measurements are obtained every 12 minutes.

As stated above the combination of these techniques can yield additional information other than the measurement of the individual concentrations. Previous measurements of OH and $HO_2+RO_2$ employed the use of a quadrupole mass spectrometer which requires measurements at specific masses (Hornbrook et al., 2011; Sjostedt et al., 2007; Mauldin et al.,



2004; Cantrell et al., 1997; Eisele et al., 1991, 1993). The current $HO_xRO_x$ system uses HR-ToF detection which allows the entire mass spectrum to be recorded for each measurement. In addition to the measurement of $H_2SO_4$ during the $NO_3^-$ mode, this feature has been used to measure numerous extremely low volatility organic compounds (ELVOCs) (Ehn et al., 2014). The present use of the HR-ToF during the OH or $HO_2+RO_2$ measurements allow one to see how these organic species

behave when the sample flow is perturbed by the addition of $SO_2$ or NO. Figure 4 shows a time series of mass 325 along with the $H_2SO_4$ signal. Mass 325 corresponds to $C_{10}H_{15}O_8(NO_3^-)$, a nitrate cluster with a common organo-peroxy radical produced from the oxidation of monoterpenes (e.g., α-pinene; Jokinen et al., 2014, Rissanen et al., 2015). These data are presented as one minute averages and include measurements from all three modes. It should be noted that the one minute averaging combines the OH and $HO_2+RO_2$ "signal" and "background" measurements such that they cannot be distinguished.

By following the $H_2SO_4$ signal, however, it is possible to see when $SO_2$ is being added or not, with the lowest values corresponding to the $NO_3^-$ ion mode where no reagents are added. From the plot it can be seen that both species are modulated and anti-correlated. The modulation arises from the instrument changing its mode of operation ($NO_3^-$, OH, or $HO_2+RO_2$). The anti-correlation indicates the conversion of $RO_2$ into $H_2SO_4$ during the $HO_2+RO_2$ portion of the measurement. By adding NO to the inlet the intermediate $RO_2$ radicals observed in the spectra should convert to

corresponding alkoxy (RO) radicals and organic nitrates ($RONO_2$; reactions 4 and 4a, respectively), with the smallest radicals giving RO and the branching steadily shifting in favor of nitrates as the molecular size increases. For the current data set, when NO was added to the sample flow, the most prominent $RO_2$'s previously observed in the α-pinene ozonolysis system (Rissanen et al. 2015) were seen to decrease, but were not removed quantitatively. One possibility could be that the NO is actually helping the autoxidation sequence to proceed by creating alkoxy intermediates which promptly isomerize and

thus advance the oxidation sequence (Kürtén et al. 2015). Figure 5 is a time series of a few selected masses along with the $H_2SO_4$ signal. From the plot it can be seen that the mass 309 and 340 peaks are anti-correlated and correlated with the addition of $SO_2$ respectively. The mass 339 peak however shows a decrease with the initial addition of $SO_2$ (OH mode), but then returns close to its non-$SO_2$ value later in the measurement series during the $HO_2+RO_2$ mode where NO is also present. The analysis of the spectra to determine the correlations and mechanisms for the formation of these species is beyond the

scope of this paper which mainly aims at providing a description of this novel combination of methods. These data are presented to demonstrate the potential extension that the combination of modes has to shed light on the chemical mechanisms for the formation of these highly oxidized organic species. In particular, the combination of the addition of NO in the $HO_2+RO_2$ mode with information from the $NO_3^-$ ion mode could elucidate formation pathways for ELVOC species proposed as crucial precursors to SOA formation. These compounds have been previously shown to form through sequential

$RO_2$ isomerization + oxygen addition reactions leading to end products with multiple carbonyl, hydroperoxy and peroxyacid functionalities (Crounse et al., 2013; Rissanen et al., 2014; Mentel et al., 2013). Some of the peroxy intermediates are directly detected by clustering with $NO_3^-$ ions (Ehn et al., 2014; Mentel et al., 2013) and thus they should also be susceptible to manipulation with the addition of gas-phase reagents. These types of analysis together with further laboratory investigations are ongoing and will be presented in future publications.



## 4 Conclusions

A new $HO_xRO_x$ instrument has been constructed which combines the previously used chemical conversion techniques for the detection of OH, and $HO_2+RO_2$ together with $NO_3^-$ ion chemical ionization and High Resolution-Time of Flight mass detection. The instrument has the ability to switch between modes which measure OH and $HO_2+RO_2$ where chemical reagents are added to the sample flow, and a $NO_3^-$ ion mode where no reagents are added to the flow. All three modes utilize HR-ToF mass detection, which, unlike previous measurements of OH or $HO_2+RO_2$, allows the entire mass spectrum to be recorded with each measurement. To the authors knowledge this is the first use of an HR-ToF for OH or $HO_2+RO_2$. Calibration coefficient values were comparable to those obtained using a quadrupole system. Similar calibration factors were measured for OH and $HO_2+RO_2$, which shows that the $HO_2+RO_2$ measurement was operated with a gain of one (i.e. one $H_2SO_4$ produce for each $HO_2$ or $RO_2$ reacted). Thus, by switching modes, an entire suite of oxidation products ($H_2SO_4$ and ELVOCs) can be measured along with the concentrations of OH and $HO_2+RO_2$ with one instrument. While the use of one instrument to obtain these individual measurements is of practical value, it is the combination of this information that extends the value beyond that of the measurements themselves. The manipulation of the inlet chemistry via the addition of gas-phase reagent species can perturb oxidation processes allowing additional information to be gained on chemical pathways and mechanisms.

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



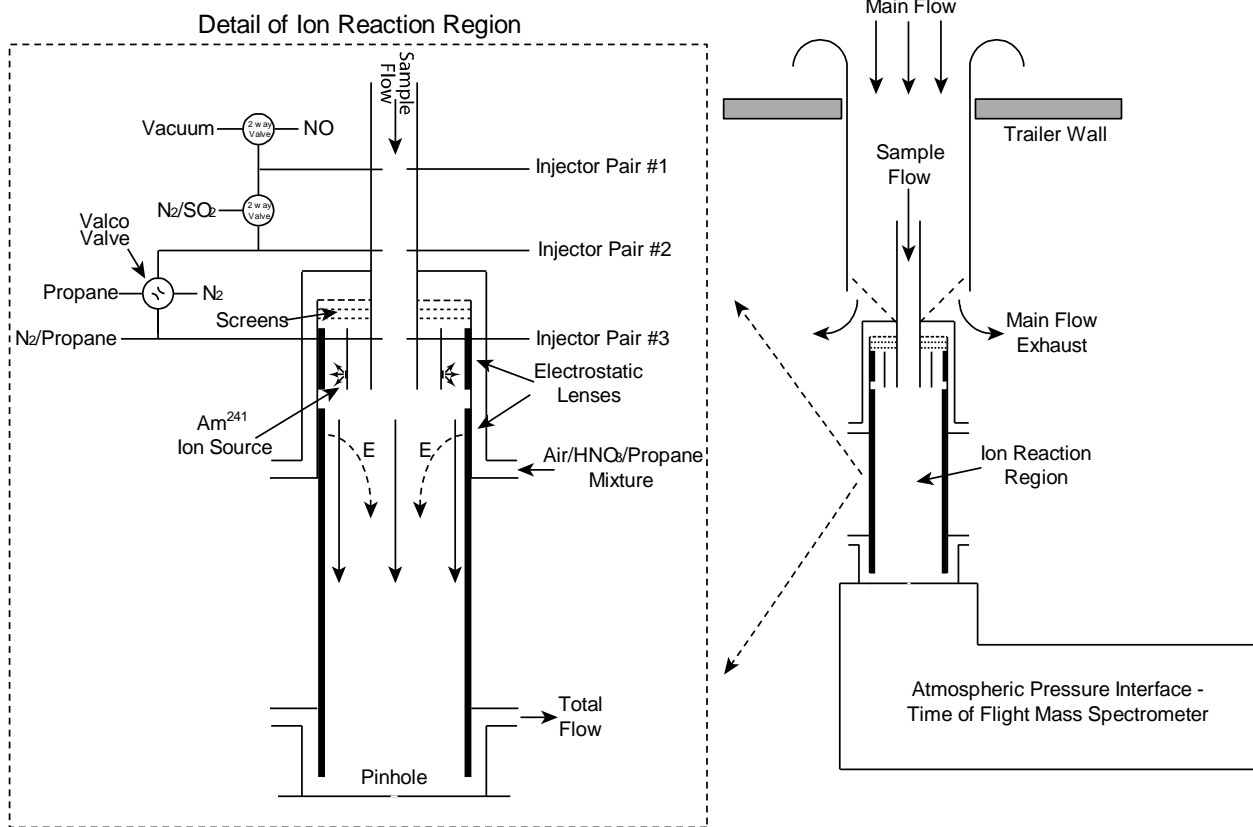

Figure 1 – Schematic of HOxROx Instrument





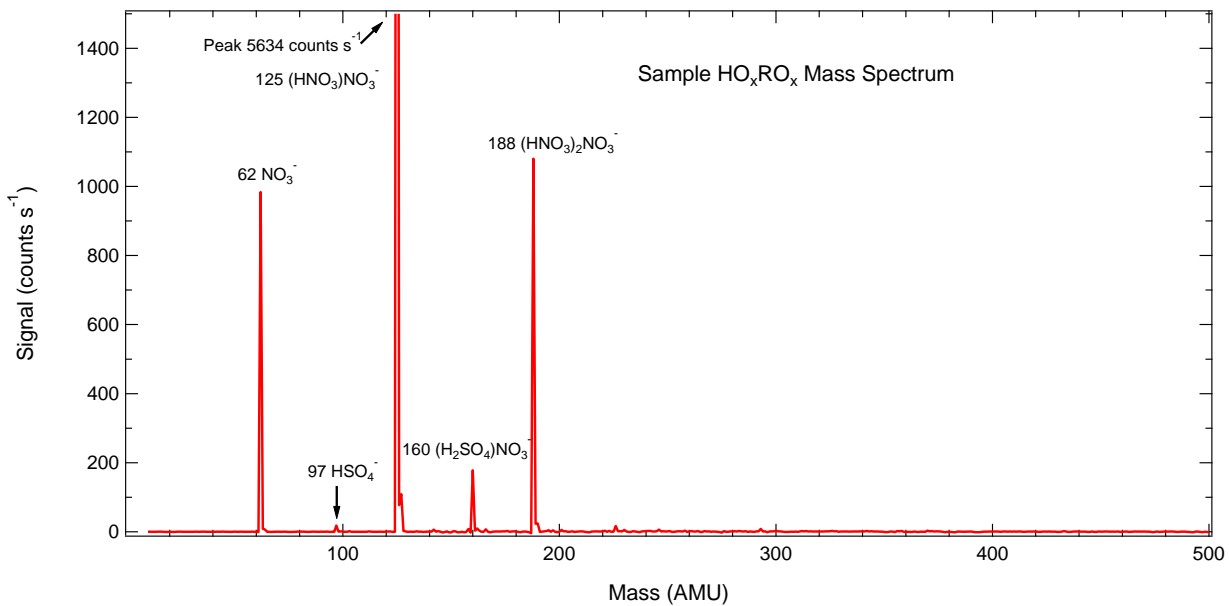

Figure 2 – Sample ambient mass spectrum obtained at the Hyytiälä research station together with peak assignments. The $HO_xRO_x$ mode is shown here to better view the product peaks, $HSO_4^-$ and $(H_2SO_4)NO_3^-$. Note that the nitrate ion peaks, $NO_3^-$, $HNO_3NO_3^-$ and $(HNO_3)_2NO_3^-$, are the dominant peaks in the spectra.





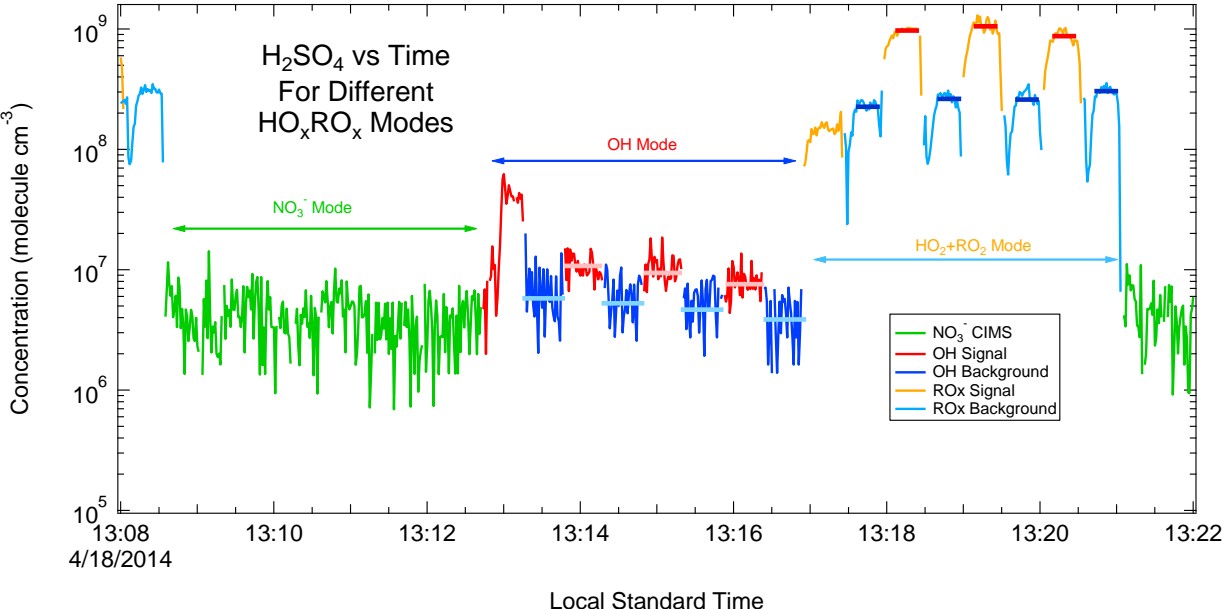

Figure 3 – Time series of $H_2SO_4$ concentrations from different HOxROx modes.





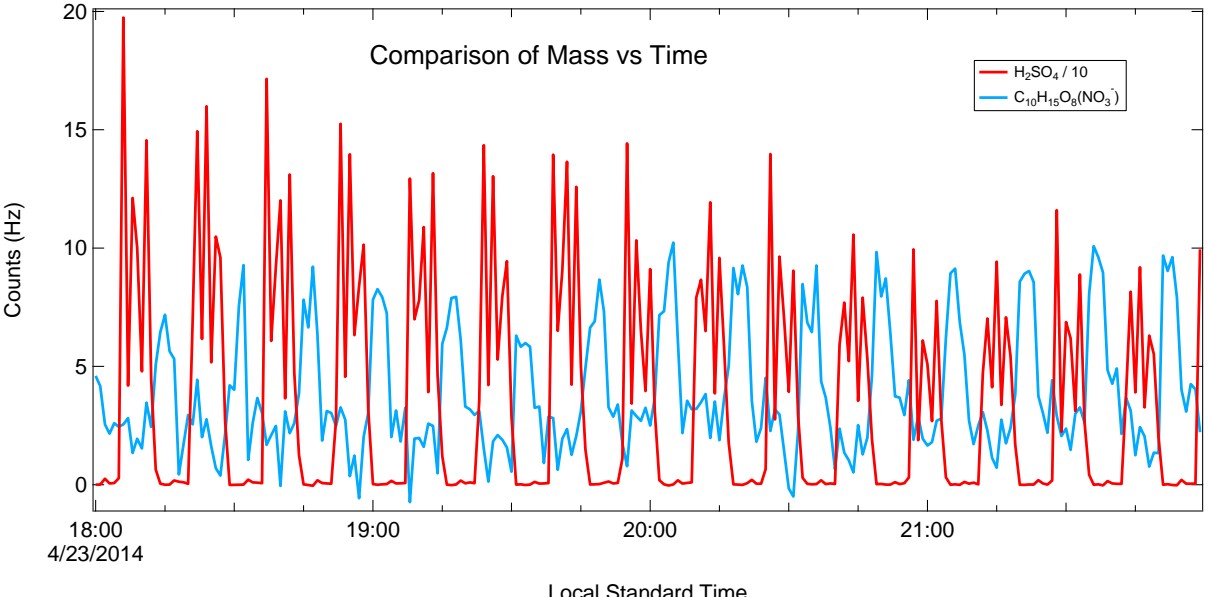

Figure 4 – Time series of $H_2SO_4$ and $C_{10}H_{15}O_8(NO_3^-)$ (mass 325). $C_{10}H_{15}O_8(NO_3^-)$ is a common organo-peroxy radical resulting from α-pinene oxidation. As can be seen, both traces are modulated and anti-correlated. The modulation arises from changes in the mode of operation of the instrument ($NO_3^-$, OH, or $HO_2+RO_2$). The anti-correlation indicates the conversion of $RO_2$ into $H_2SO_4$ during the $HO_2+RO_2$ portion of the measurement.





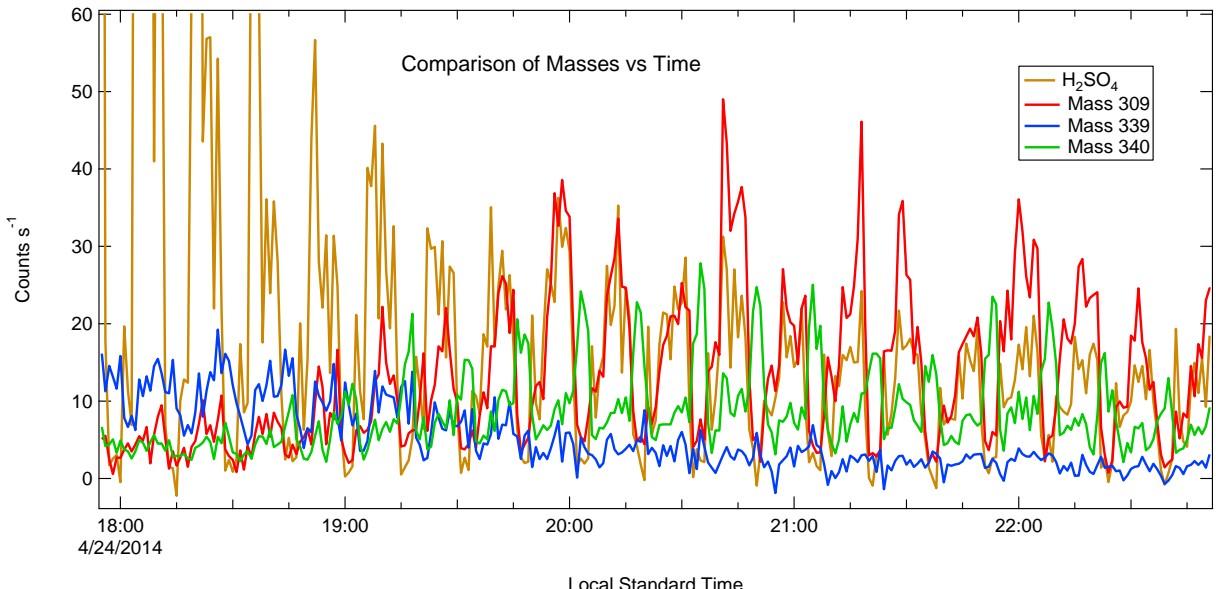

Figure 5 – Time series of $H_2SO_4$ and other selected masses obtained with the $HO_xRO_x$ instrument.

