# Peer review of "Furthering information from OH and HO2+RO2 observations using a high resolution time of flight mass spectrometer"

_Atmospheric Measurement Techniques, 2015_

## Referee Comment (RC1) · Anonymous Referee #1 · 15 Mar 2016

Review of "Furthering information from OH and $HO_2+RO_2$ observations using a high resolution time of flight mass spectrometer"

This manuscript is an instrument paper describing the advancement of a typical $HO_xRO_x$-CIMS instrument (i.e. Sjostedt et al 2007) by upgrading from a quadrupole mass spectrometer to a Time of Flight instrument (CI-APi-ToF). This change is mass spectrometer is a subtle yet important advancement for this type of instrument. Problems with duty cycle (the necessity of mass "hopping" with a quadrupole) are removed and the entire mass spectrum can be measured with each acquisition. This provides additional information previously unavailable (or at least very difficult to obtain) with quadrupole instruments. The authors have shown a small amount of data at the end of the paper showing how the addition of $SO_2$, NO and propane (the OH scrubber) for the $HO_x$ measurements perturbs the measurements of more large highly oxidized organic molecules (ELVOC, HOM's). It would be nice to see this explored further (perhaps a part 2 of this paper)? The paper is clearly written and fits well into the scope of AMT. I recommend the paper should be published subject to a few minor comments below.

General Comments:

While the descriptions of the operating modes of the instrument ($NO_3^-$, OH, OH-Background, $HO_2+RO_2$, and $HO_2+RO_2$ background are detailed well, a little more detail about the operating conditions of the mass spectrometer would be appreciated (i.e. SSQ pressures, field strength used across the sheath/total flow lenses, ToF extraction frequency). While nitrate sources have been previously used on CI-APi-ToF instruments (i.e. Ehn et al 2014 and references therein), I believe a detailed account of the operating conditions is important as the $HO_xRO_x$ front end does differ from the standard $NO_3^-$ front end. Also how is the sheath air "filtered", (charcoal scrubber?)? While backgrounds for OH and $HO_2+RO_2$ are discussed at length how is the background measurement for $H_2SO_4$ performed? Is simply inferred that the background count rates are 0 at masses 97 and 160 when there is no $H_2SO_4$ present? Presumably this background could rise if the sheath air wasn't being scrubbed of adequately of ambient $SO_2$.

Specific Comments:

P2L30: The author's refer to the mass spectrometer used in this instrument as an HR ToF while previously having called it a CI-APi-ToF. Please be consistent with the terminology so that those not in this community are not confused as to the mass spectrometer being used.

P3L20: Sjostedt et al also added $HNO_3$ through the CIMS rear injectors in addition to that added to the sheath gas to maintain the $HNO_3$ cluster distribution in the instrument. Was this found to be unnecessary with this particular inlet geometry? Does the cluster distribution change over time or even between operating modes? Does that effect the sensitivity? This is the type of information not available with a quadrupole system with a collisional dissociation chamber (CDC) designed to strip the clusters down to bare ions and certainly should be exploited with the ToF.

P4L17: The authors note that $H_2SO_4$ is detected at masses 97 and 160. Is the sum of the two used for quantification or only one of them? Please be clear.

P6L24: Are these detection limits for a 1 second measurement or has the data been averaged in post processing?

P7L14: The OH backgrounds shown in Figure 3 are high, due to the measurement being performed with unlabeled $SO_2$ instead of $^{34}SO_2$. I'm curious if the authors have a feeling about by how much the detection limit would be lowered (likely) by not having the measurement sit on top of a varying $H_2SO_4$ background. Of course whether this increase would be worth (the rather substantial) cost increase of using $^{34}SO_2$ would be debatable.

P8L15: Should be reactions R5 and R5a

P15: Figure 4. Since the data presented are 1 minute averages I think it would be useful to show the reader the standard deviation of the measurement so they can get a feel for the point to point variability as the count rates for the $C_{10}H_{15}O_8(NO_3^-)$ cluster are very low. In fact it's probably unnecessary to show 4 hours' worth of data. Displaying a smaller chunk of data would make the plot easier to read while still making the point that the $C_{10}H_{15}O_8(NO_3^-)$ cluster is anti-correlated with $H_2SO_4$.

P16: Figure 5: The same comment as above. It might be useful to stack the time series vertically as opposed to simply overlaying them on top of each other. The authors could perhaps put some type of shading in the background of the figure to denote when the instrument is switching between different modes.

References:

Ehn, M., et al., Mass spectrum of ELVOCs produce by a-pinene ozonolysis, Nature, 506, 476-479, doi:10.1038/nature 13032, 2014.

Sjostedt, S. J., Huey, L. G., Tanner, D. J., Peischl, J., Chen, G., Dibb, J. E., Lefer, B., Hutterli, M. A., Beyersdorf, A. J., Blake, N. J., Blake, D. R., Sueper, D., Ryerson, T., Burkhart, J., Stohl, A., Observations of hydroxyl and the sum of peroxy radicals at Summit, Greenland during summer 2003, Atmos. Environ., 41, 5122–5137, 15 doi:10.1016/j.atmosenv.2006.06.065, 2007.

---

## Referee Comment (RC2) · Anonymous Referee #2 · 24 Mar 2016

This manuscript describes the coupling of previously developed OH, H2SO4, HO2+RO2, and oxidized organics measurements using a High Resolution Time-of-Flight Mass Spectrometer. The system has the same sensitivity towards OH and HO2+RO2 as previous quadrupole CIMS measurements. Some sample data from measurements conducted at the Hyytiälä research station were provided. The authors proposed that the combination of these measurements into one instrument can provide additional insights into the formation mechanisms of organic species.

The manuscript is generally well-written. The authors described the method thoroughly and took great care to obtain appropriate backgrounds and limit the additional cycling of radicals from contributing to the HSO4- signal. The measurements described are

very useful on their own, and the practical gains from being able to perform them using a single instrument can be significant.

However, I do not think that the manuscript in its current form suffices as a standalone publication. While the coupling of all measurements is new, all the measurement techniques described in the manuscript are the same as what have been done previously for quadrupole CIMS, and OH, H2SO4, and HO2+RO2 measurements have also been conducted together before. In this regard, the "instrumentation" section reads like a summary of previously CIMS literature. The addition in this manuscript is the monitoring of oxidized organic species which is made possible by the collection of complete mass spectra on the HR-ToF-CIMS, something that is not possible with a quadrupole. Nevertheless, the use of NO3- with HR-ToF-CIMS is not new and has already been the topic of multiple recent publications, which are cited in the manuscript.

The title is also misleading in that no information is presented here which utilizes OH or HO2+RO2 measurements, but rather the addition of NO or SO2. The authors claimed that addition of NO and SO2 may provide insights into oxidation mechanisms, and pointed out observations of representative masses for the a-pinene system. The SO2-induced decrease in some of the signals (during the OH mode) is not explained. The effect of NO is due to the titration of oxidized RO2. The incomplete decrease in RO2 when adding NO is thought to be due to the formation of an alkoxy radical which can isomerize and aid in the oxidation sequence, but this is speculative. The authors noted that the purpose of the current manuscript is not to analyze the changes in the spectra and that this is the topic of future work, but more support should be provided from other masses in the spectra to showcase the usefulness of perturbations, which are limited in NO concentration by the need to measure HO2+RO2.

Overall, while the coupling of all measurements is new, the descriptions of the various measurement modes are essentially the same as previous CIMS literature, the combined system also has the same sensitivity towards OH and HO2+RO2 as previous CIMS measurements. I agree with the authors that the combined system could potentially provide additional insights into the oxidation mechanisms, however, this is not demonstrated sufficiently in this manuscript. With all this, I do not think this manuscript adds substantially to literature as it stands. It is clear that the authors have already acquired ambient data from Hyytiälä research station with the combined instrument. I do not think that there is a need to separate their work into two papers, one on instrumentation and one on the data analysis, since all the measurement techniques are well-established already. A manuscript with some brief descriptions of the measurement techniques and in-depth data analysis to showcase the potential capability of the combined system in providing further insights into oxidation mechanisms, on the other hand, would have been much stronger and would be of great interest, but will be more suitable for a more general journal (not an instrumentation journal).

Specific/minor comments:

Figure 2: The absolute (HNO3)NO3- counts seem low. Is this because of transmission or long reaction time in the inlet? If reaction time is long, are the authors concerned about secondary reactions of HSO4- with oxidized organics? Does this affect the behavior of the organic signals from perturbations?

Page 2 line 6: "mass spectroscopy" should read "mass spectrometry"

Page 5 line 10: "forming organo-nitrates through reaction 4a", should refer to 5a.

Page 6 lines 25-27: Breakdown of errors would be useful.

Page 7 line 17-18: "The difference between the OH "signal" and "background" values is the concentration of H2SO4 attribute to production from OH", should change "attribute" to "attributed".

Page 8, line 15: "(RONO2; reactions 4 and 4a, respectively" should read 5 and 5a.

Page 8, lines 21-22. "From the plot it can be seen that the mass 309 and 340 peaks are anti-correlated and correlated with the addition of SO2 respectively." Mass 309 is correlated and 340 is anti-correlated.

Figure 1. The 3 in HNO3 needs reformatting in the diagram.

Figure 3. I suggest using multiple y-axis, log-linear scale can be used if necessary. The log scale makes time series unclear. If this is reformatted, lines pointing out H2SO4 concentrations are probably unnecessary.

---

## Author Comment (AC1) · 22 Apr 2016

Reply to referees for manuscript amt-2015-398 "Furthering information from OH and HO2+RO2 observations using a high resolution time of flight mass spectrometer" by Mauldin et al. We would like to thank the referees for their time to review the manuscript and their comments. We will respond to them below.

Referee #1 General Comments These comments refer mainly to a need of more description of the operating conditions of the instrument. These comments arise out of a misunderstanding that the HOxROx chemical ionization source is different from those used in previous studies. In fact the source used in this study as well as other nitrate CIMS sources used recently for TOF measurements are all based upon the same design and dimensions as those used originally by Eisele. Text to this fact has been added to the NO3- Mode section. Specific Comments The term HR-ToF has been replaced with CI-APi-ToF as requested Question as to whether we also add HNO through the rear injectors as Sjostedt et al. did. We also do. This maintains the reagent ion (NO3-) cluster distribution which is important for keeping the instrument calibration constant. Text has been added to the OH Mode section to this effect. Question as to whether we use the sum of the signals at 97 Th and 160 Th for the calculation of H2SO4. Yes, and text has been added to the NO3- Mode section to this effect. Question regarding the stated detection limits. The stated limits are for the entire signal and background measurements. Text clarifying the time has been added. Question as to how using unlabeled SO2 changes the detection limit. Our feeling is not much. While the background is increased, it only perhaps doubles from that when labeled SO2 is used. The largest effect are changes in the H2SO4 between signal and background measurements, but this effect goes down as the OH goes down. Labeled is always preferable. We changed the wrongly cited reactions 4 and 4a to 5 and 5a. The final comments were towards making Figures 4 and 5 more readable. We expanded both plots to show less time as suggested.

Referee #2 While the coupling of all measurements is new, all the measurement techniques described in the manuscript are the same as what have been done previously for quadrupole CIMS, and OH, H2SO4, and HO2+RO2 measurements have also been conducted together before. In this regard, the "instrumentation" section reads like a summary of previously CIMS literature. While all of the techniques have been previously reported we believe the summary given aids the reader, especially in regards to the chemistry of the different modes when discussing the organic species. The addition in this manuscript is the monitoring of oxidized organic species which is made possible by the collection of complete mass spectra on the HR-ToF-CIMS, something that is not possible with a quadrupole. Nevertheless, the use of NO3- with HR-ToF-CIMS is not new and has already been the topic of multiple recent publications, which are cited in the manuscript. Yes, the NO3- has found wide use in VOC oxidation experiments currently due to its selectivity toward highly-oxidized species detection, but it is the combination of the ability to see these species together with the quantification of their source strengths that is the novelty here. The authors claimed that addition of NO and $SO_2$ may provide insights into oxidation mechanisms, and pointed out observations of representative masses for the a-pinene system. The $SO_2$- induced decrease in some of the signals (during the OH mode) is not explained. A section has been added to address this comment. The authors noted that the purpose of the current manuscript is not to analyze the changes in the spectra and that this is the topic of future work, but more support should be provided from other masses in the spectra to showcase the usefulness of perturbations, which are limited in NO concentration by the need to measure $HO_2+RO_2$. We have changed the mass spectrum figure to show spectra from two different modes ($NO_3$- and $HO_2+RO_2$) as well as expanded versions showing the behavior of the organic species in these modes. Discussion of this figure and the changes between modes has been added to the text. I do not think that there is a need to separate their work into two papers, one on instrumentation and one on the data analysis, since all the measurement techniques are well-established already. A manuscript with some brief descriptions of the measurement techniques and in-depth data analysis to showcase the potential capability of the combined system in providing further insights into oxidation mechanisms, on the other hand, would have been much stronger and would be of great interest, but will be more suitable for a more general journal (not an instrumentation journal). While we recognize that much of the information gathered on these measurement and chemical conversion techniques have been presented previously, we find it very useful to have it all described in one publication due to, for example, poor availability of the previous papers for wider audience and the scattered nature of the previous reports. So even though we noticed this redundancy, we still thought the gain of this clearly outweighs the potential harm of repetition (it should also be noted that we definitely do want to cite all the relevant papers where progress has been made in these techniques). The quantitative data interpretation from the field with these type of chemical perturbation experiments is

a very tedious task due to large amount of peaks in the ToF spectra, which origin is currently unclear. This is true especially with this type of technique (NO3- ionization), which only measures selected highly-oxidized end-products of an oxidation chain reaction with large uncertainties still existing in their specific formation mechanisms. The "full interpretation" of the spectra will take much more time and potentially require specific lab experiments to support the identifications. Thereby we feel that the advantage of using these techniques together is significant and merits a separate publication outlining this to a wider audience. As a result we want to keep the discussion of the details of the techniques largely as it is now. Nevertheless, due to the Referee's comment the discussion was extended and Figure 3 augmented to more clearly illustrate the usefulness of this combination of techniques. Specific Comments Question as to the count rate for the reagent ion seeming low. This value is well within operational values for the measurement and is similar to that seen by other NO3- ToFs in our group. Spectroscopy is now spectrometry. 4a is now 5a in the text A discussion of the error and times has been given Attribute is now attributed 4 and 4a have been changed to 5 and 5a Figure 3 has been changed

Again we would like to thank the referees for their helpful comments and suggestions. They have made the manuscript a better piece of work.

Please also note the supplement to this comment:
http://www.atmos-meas-tech-discuss.net/amt-2015-398/amt-2015-398-AC1-supplement.pdf

**Supplement:**

[revised manuscript text omitted]

Figure 3 show typical mass spectra obtained from the HOxROx instrument at the Hyytiälä research station. The major peaks have been labeled with their mass and the compounds they represent. The spectrum in Figure 3a was obtained while the instrument was in the $HO_2+RO_2$ "signal" mode to better view the $HSO_4^-$ (mass 97) and $(H2SO¬4)NO_3^-$ (mass 160) product peaks. The high resolution of the CI-APi-ToF allows these product ions to be detected at their exact masses, 96.9601 amu and 159.9557 amu ensuring that they indeed were $HSO_4^-$ and $(H_2SO_4)NO_3^-$ respectively. It can be seen that the reagent nitrate ion peaks; $NO_3^-$, $(HNO_3)NO_3^-$, and $(HNO_3)_2NO_3^-$, are the dominant peaks in the spectra. Even though the spectrum shown was obtained in the $HO_2+RO_2$ "signal" mode which produces the most product $H2SO4$, the reagent ion still comprises ~97% of the total ion signal. The inset in the Figure illustrates the highly-oxidized products detected, with a few of the major identified product species labeled by their corresponding masses.

In Figure 3b a mass spectrum obtained while in the $NO_3^-$ mode is presented. In this mode no reagents are added to the sample flow, which can be directly seen by the absence of the large product peaks at 97 and 160 Th. Also differences in the highly-oxidized product distribution at the higher masses are apparent and are illustrated in the inset of the Figure (i.e., compare the labelled peaks between the Figures a and b). Note especially the lower signal strengths of the masses 309 and 339 Th which have been previously identified as prominent organonitrates (Kulmala et al 2013). Both of these signals go up by addition of NO to the sample gas mixture, with a clear decrease in prominent radical signals (e.g., at 325 Th and 357 Th assumed to originate from MT ozonolysis, see Rissanen et al. 2015), indicating organonitrate formation through 5a and thus supporting the previous identification. Also the $SO_2$ addition is seen to modulate some of the HOM signals, but the reason for this behavior is currently unclear. In Figure 4, a time series of $H_2SO_4$ together with other selected masses is shown. Larger values of $H_2SO_4$ are indicative of $SO_2$ addition from either the OH or $HO_2+RO_2$ measurement modes. From the plot it can be seen that the mass 309 is correlated and mass 340 is anti-correlated with the addition of $SO_2$ respectively. The mass 339 peak however shows a decrease with the initial addition of $SO_2$ (OH mode), but then returns close to its non- $SO_2$ value later in the measurement series during the $HO_2+RO_2$ mode where NO is also present. This modulation could be explained by gas-phase organosulfur compound formation, potentially organoulfonates or organosulfates, but the mechanism behind this is presently unknown, as the organosulfur compounds are generally thought to originate from condensed phase reactions (Riva et al., 2016; Kramer et al., 2016).

In Figure 5 a time series of mass 325 along with the $H_2SO_4$ signal is shown. Mass 325 is mainly assumed to originate from a $C_{10}H_{15}O_8(NO_3^-)$ species - a nitrate cluster with a common organo-peroxy radical produced from the oxidation of

monoterpenes (e.g., α-pinene; Jokinen et al., 2014, Rissanen et al., 2015). These data are presented as one minute averages and include measurements from all three modes. It should be noted that the one minute averaging combines the OH and $HO_2+RO_2$ "signal" and "background" measurements such that they cannot be distinguished. By following the $H_2SO_4$ signal, however, it is possible to see when $SO_2$ is being added or not, with the lowest values corresponding to the NO3- ion mode

5    where no reagents are added. From the plot it is evident that both species are indeed modulated and anti-correlate with each other. The modulation arises from the instrument changing its mode of operation ($NO_3^-$, OH, or $HO_2+RO_2$). The anti-correlation indicates the conversion of $RO_2$ into $H_2SO_4$ during the $HO_2+RO_2$ portion of the measurement. By adding NO to the inlet the intermediate $RO_2$ radicals observed in the spectra should convert to corresponding alkoxy (RO) radicals and organic nitrates ($RONO_2$; reactions 5 and 5a, respectively), with the smallest radicals giving RO and the branching steadily

10   shifting in favor of nitrates as the molecular size increases. For the current ambient data set, the addition of NO lead to significant decreases in the $RO_2$'s detected, but did not remove them quantitatively (Figure 5). One possible reason for this behavior could be that the NO addition is actually helping certain autoxidation pathways to proceed by creating alkoxy intermediates which promptly isomerize and thus advance the oxidation sequence (Kürtén et al. 2015, Mentel et al. 2015), and result in isomeric peroxy radical products detected at the same masses in the spectra.

15   The combination of quantifying the amount of direct (=OH) and indirect (=$HO_2+RO_2$) oxidants together with their specific low-volatile reaction products can be highly useful for gaining insight into atmospheric oxidation pathways, especially relating to SOA formation. The simultaneous quantification of the variation in the main day-time oxidizer, the OH radical, with the variation in the formation rate of highly-oxidized molecules will inform about the oxidation pathways, but also it can improve our knowledge on the limitations of this processes, e.g., is the HOM formation oxidant and/or VOC limited in

20   the environment under considerations by observing the correlation of these concentrations. While data in Figures 4 and 5 show clear correlations in the changes of the oxidized organics with the addition of reagent gasses, detailed analysis of the spectra to determine the correlations and mechanisms for the formation of these species is beyond the scope of this paper which mainly aims at providing a description of the advantage of using this novel combination of methods. These data are presented to demonstrate the potential extension that the combination of modes has to shed light on the chemical

[revised manuscript text omitted]